# Insight into the Structural Dynamics of the Lysenin During Prepore-to-Pore Transition Using Hydrogen–Deuterium Exchange Mass Spectrometry

**DOI:** 10.3390/toxins11080462

**Published:** 2019-08-07

**Authors:** Magdalena Kulma, Michał Dadlez, Katarzyna Kwiatkowska

**Affiliations:** 1Institute of Biochemistry and Biophysics, Polish Academy of Sciences, 5A Pawinskiego St., 02-106 Warsaw, Poland; 2The Nencki Institute of Experimental Biology, Polish Academy of Sciences, 3 Pasteur St., 02-093 Warsaw, Poland

**Keywords:** hydrogen-deuterium exchange, lysenin, pore formation, structure, toxin

## Abstract

Lysenin is a pore-forming toxin of the aerolysin family, which is derived from coelomic fluid of the earthworm *Eisenia fetida*. Upon binding to sphingomyelin (SM)-containing membranes, lysenin undergoes a series of structural changes promoting the conversion of water-soluble monomers into oligomers, leading to its insertion into the membrane and the formation of a lytic β-barrel pore. The soluble monomer and transmembrane pore structures were recently described, but the underlying structural details of oligomerization remain unclear. To investigate the molecular mechanisms controlling the conformational rearrangements accompanying pore formation, we compared the hydrogen–deuterium exchange pattern between lysenin^WT^ and its mutant lysenin^V88C/Y131C^. This mutation arrests lysenin oligomers in the prepore state at the membrane surface and does not affect the structural dynamics of the water-soluble form of lysenin. In contrast, membrane-bound lysenin^V88C/Y131C^ exhibited increased structural stabilization, especially within the twisted β-sheet of the N-terminal domain. We demonstrated that the structural stabilization of the lysenin prepore started at the site of lysenin’s initial interaction with the lipid membrane and was transmitted to the twisted β-sheet of the N-terminal domain, and that lysenin^V88C/Y131C^ was arrested in this conformation. In lysenin^WT^, stabilization of these regions drove the conformational changes necessary for pore formation.

## 1. Introduction

Lysenin is a toxic protein produced by *Eisenia fetida* that plays important roles in this earthworm’s innate immunity and defense strategies against parasites [1,2]. Lysenin belongs to the family of aerolysin-like pore-forming proteins (aβ-PFPs), of which the founding member (aerolysin) is a well-studied toxin produced by the pathogenic bacterium *Aeromonas hydrophila*. The aβ-PFP family includes numerous toxins present throughout all kingdoms of life: bacteria, fungi, plants, and animals [3,4,5,6,7,8,9]. While many aβ-PFPs family members produced by pathogenic bacteria (e.g., *Aeromonas*, *Clostridium*, and *Bacillus*) act as virulence factors in food poisoning and infections, aβ-PFP family members produced by eukaryotes serve in defense against pathogens or parasites [6,10,11]. The amino acid sequences of all aerolysin-like proteins exhibit low homology but share a similar monomeric architecture, particularly a conserved N-terminal domain involved in β-barrel pore formation.

Like most pore-forming toxins, lysenin is produced and secreted as a soluble monomer. Crystallographic analyses reveal that lysenin comprises two main structural domains: an N-terminus pore-forming module (PFM) at amino acid position 1–160, which contains a SM-binding site and the membrane insertion loop (position 34–107), and the C-terminal β-trefoil receptor-binding domain (RBD) at amino acid position 161–297, which is responsible for initial binding with the lipid membrane [9] (Figure 1A). The crystal structure of the monomeric form of lysenin bound to SM reveals that toxin binding requires both the hydrophilic (phosphocholine head group; POC) and hydrophobic (acyl chain tail) lipid portions [9] (Figure 1B). Previous crystallographic studies and analyses of the binding of deletion mutants have revealed that lysenin interacts with membranes in a two-stage process [9,12]. First, a patch of positively charged amino acids in the C-terminal domain attracts toxin to the membrane surface and allows the initial binding of the C-terminal domain to the POC of SM via both a hydrogen bond network (including S227, Y233, and Y282) and a salt bridge interaction (K185). Second, two critical tyrosine residues (Y24 and Y26) located in the PFM of the N-terminal domain bind the acyl chain of SM via ring stacking–like interactions [9]. Detailed analysis of the crystal structure of the lysenin/SM complex has revealed that other PFM residues, including K21, Q117, and E128, also interact with the SM head group, creating an additional hydrophilic binding site. The multivalent SM binding site at the C-terminal domain facilitates the interaction of lysenin with clustered SM organized into the SM-rich domains of the plasma membrane, thus increasing the probability of a monomer-monomer encounter [13,14,15,16,17]. In turn, this enables lysenin monomers to create oligomeric structures (prepores) on the lipid membrane surface, which undergo a large conformational rearrangement, leading to insertion into the lipid membrane and pore formation.

Blue native electrophoresis has revealed that stable oligomer formation in SM-containing membranes correlates with lysenin self-assembly into trimers that act as the functional unit of the protein bound to SM [13,14]. Lysenin oligomerization, but not binding, is facilitated by cholesterol presence, which increases the fluidity of SM-containing membranes and promotes separation of the SM-rich liquid-ordered phase [14,17,18,19]. Previous studies based on electron microscopy, atomic force microscopy (AFM), and two-dimensional electron crystallography have revealed that lysenin oligomers assemble into a hexagonal close-packed (hcp) structure with an external diameter of ~10–11 nm and an inner pore of ~6 nm [9,16,19]. AFM measurements of hcp assemblies of lysenin on a SM-containing bilayer have elucidated the presence of two oligomer populations with different heights—2.6 and 5 nm. Investigations of the cholesterol-dependent cytolysin perfringolysin O (PFO) have also revealed the presence of two distinct oligomer populations, while the lower-height oligomers (2.6 nm) correspond to membrane-inserted oligomers (pores), and the taller oligomers (5 nm) represent the prepore state [20,21]. The height of the putative lysenin’s prepore corresponds to the height of the water-soluble lysenin monomer, suggesting that lysenin protomers in the prepore state do not undergo radical structural rearrangements that change the protein’s orientation to the membrane surface. 

The conversion from the prepore to the pore state entails critical conformational reorganizations of lysenin structure, which only involve the N-terminal domain of lysenin (Figure 1C). It has been suggested that the contacts between neighboring monomers in the prepore state, and the geometrically and energetically conductive intramolecular interactions between β-strands, are triggers for displacement of the insertion loop covering the region from amino acid residues 44–67 (the tongue). Insertion loop movement leads to breakage of the hydrogen bonds between the β-strand pairs β3 (pos. 38–44) and β9 (pos. 125–128), and β5 (pos. 68–81) and β11 (pos. 147–156), as well as displacement of the strands β5 and β6 (pos. 84–90). This leads to insertion loop unfolding, bringing together the parallel-oriented strands β2 (pos. 12–27) and β10 (pos.129–138) of adjacent monomers, and tilting the twisted β-sheet containing the strands β2, β8 (pos. 111–121), β10, and β11 by a 45-degree angle relative to the receptor-binding domain. The reorganization of the N-terminal domain within the strands β4 (pos. 54–57), β5, and β6 (pos. 84–90), as well as a single α3_10_ helix (pos. 91–96) of each protomer, forces the extending strands β3 and β7 (pos. 99–103) to reassemble into tilted and twisted β-hairpins (amino acid residues 34–107). Finally, nine β-hairpins curve nearly 270° during incorporation into the lipid membrane, constructing a final β-barrel structure. Overall, the lysenin pore exhibits a mushroom-like architecture with the β-barrel creating a central stem passing through the lipid membrane, and the mushroom cap formed by the C-terminal receptor-binding domains (bottom part of the cap) and β-strands (β2, β8, β10, and β11) originating from the N-terminal domains (upper part of the cap, collar) (Figure 1D). 

The transition of a toxin from a water-soluble form to a membrane-anchored form requires dramatic structural reorganization. Although recent studies have elucidated several aspects of pore formation by lysenin [8,22], the structural characterization of intermediate steps of the pore-formation pathway remains incomplete. To gain insight into the structural dynamics of lysenin in the pre-pore state, and to characterize structural changes accompanying its insertion into the lipid membrane, we applied hydrogen–deuterium exchange mass spectrometry (HDX-MS).

HDX is a complex process whose kinetics depends on both protein dynamics and intrinsic amide hydrogen exchange rates. Two factors may affect the hydrogen–deuterium exchange: solvent accessibility and H-bonding. However, water molecules penetrate protein structures quite efficiently. Therefore, it is believed the main factor protecting from exchange is the involvement of amide in H-bonding, i.e., the stability of local secondary structure elements, while solvent accessibility has minor influence [23]. Generally, due to the fact that protection against hydrogen–deuterium exchange may be best correlated with the “order” in a given region of protein, we interpret the results in terms of structural stability in a given region. Therefore, this method provides useful information regarding the dynamics of structural elements of pore-forming proteins, both in solution and upon interaction with a lipid membrane and enables investigation of the conformational changes accompanying transitions between protein states with high reproducibility and precision [24,25,26].

Previous studies have indicated that introducing the double-cysteine mutation V88C/Y131C in lysenin enables creation of a disulfide bond that locks together the β6 and β10 strands and prevents β6 displacement from the rest of the toxin molecule. Consequently, this mutation abolishes lytic activity while preserving oligomer formation [22], such that lysenin^V88C/Y131C^, is an ideal candidate for characterizing the structural dynamics of lysenin oligomers arrested in the pre-pore state. To obtain structural information regarding the mechanism of lysenin oligomerization, we compared the patterns of the exchange of amide protons in lysenin^WT^ with those in lysenin^V88C/Y131C^, both in aqueous solution and upon binding to SM-containing membranes. This approach enabled mapping of the regions in the lysenin molecule where the kinetics of hydrogen–deuterium exchange are substantially altered after interaction with the model lipid membranes.

Our present results indicate the structural dynamics of the lysenin regions involved in the two stages of pore formation: transition of the water-soluble lysenin monomer into the ring-shaped oligomer and its conversion into the β-barrel membrane-inserted pore. We demonstrated that the double-cysteine mutation promoted stabilization of the majority of lysenin regions upon binding the lipid membrane. Moreover, the structural stabilization of lysenin^V88C/Y131C^ was more pronounced in the regions that form the N-terminal cap domain in the lysenin pore state, which have not previously been considered to be crucial in the oligomerization and pore formation processes. These findings suggest that increased stability in these regions supports the maintenance of lysenin in the correct orientation relative to the membrane and adjacent monomers in the pre-pore complex and forces further structural rearrangements that drive pore formation.

## 2. Results

### 2.1. Lysenin^V88C/Y181C^ Binding to SM-Containing Membranes

The first step in transmembrane pore formation is the binding of lysenin to SM of the lipid bilayer. The crystal structure of lysenin bound to SM reveals that two sites are responsible for SM binding: the positively-charged patch on the C-terminal domain and the tyrosine residues (Y24 and Y26) in the β2 strand of the PFM in the N-terminal domain [9]. The cysteine mutations V88C and Y131C are introduced in the β6 and β10 strands, outside of the region responsible for the interaction between lysenin and SM. However, studies of perfringolysin O (PFO), a member of the cholesterol-dependent cytolysin family of pore-forming toxins, have shown that modification within the pore-forming domain alters the PFO’s interaction with the lipid membrane, indicating structural and functional coupling between two distant and spatially separated domains of the toxin [27]. Lysenin^V88C/Y131C^ exhibits disabled lytic activity [22], but its interaction with SM in the membrane has not previously been characterized. In the present study, a lipid overlay assay revealed that both lysenin^WT^ and lysenin^V88C/Y131C^ selectively recognized and bound SM in a dose-dependent manner, starting from 5 pmol lipid, and with high specificity (not recognizing cholesterol or phospholipids, such as DOPC or DPPC), indicating that lysenin^V88C/Y181C^ retained the selectivity for SM binding that was observed in lysenin^WT^ (Figure 2A).

We also demonstrated the ability of lysenin^V88C/Y181C^ to bind to SM based on surface plasmon resonance measurements using liposomes composed of SM mixed with DOPC and cholesterol (molar ratio 1:2:1). Previous studies show that, when oxidized, the double-cysteine mutation V88C/Y131C retains disulfide-trapped lysenin oligomers on the membrane surface upon binding to a SM-containing bilayer. However, when a reducing agent (e.g., DTT) breaks the disulfide bond between these cysteine residues, lysenin oligomers become inserted into the lipid bilayer [22].

Sensogram analysis revealed that lysenin^WT^ reached a plateau value of ~6900 RU after a dissociation time of 300 s. In comparison, lysenin^V88C/Y181C^ displayed a stronger tendency for dissociation, reaching 5300 RU at this time-point (Figure 2B). In contrast, the dissociation curves of lysenin^V88C/Y181C^ in the presence of 10 mM DTT were comparable to those of lysenin^WT^, reaching a value of ~6800 RU after a dissociation time of 300 s (Figure 2B).

To further analyze the interactions of lysenin^WT^ and lysenin^V88C/Y131C^ with the lipid membranes, we measured changes in the surface pressure of the lipid monolayer. The surface-active properties of lysenin molecules were initially characterized by measuring spontaneous adsorption of the proteins at the argon–water interface. Injection of both lysenin^WT^ and lysenin^V88C/Y131C^ into the aqueous sub-phase prompted a two-phase increase of surface pressure in a dose-dependent manner (Appendix A). Adsorption kinetics analysis showed that, when using lower protein concentrations of 0.5–2 nM, the rates of surface pressure changes were higher with lysenin^V88C/Y131C^ than with lysenin^WT^. However, at higher protein concentrations, both proteins exhibited similar kinetics of surface pressure changes, reaching surface pressure saturation (16 mN/m) at a concentration of 40 nM (Appendix A).

To initiate surface pressure changes in mixed lipid monomolecular layers spread at the argon–water interface, we injected 40 nM lysenin^WT^ and lysenin^V88C/Y131C^ into the buffer sub-phase. For the monolayer composed of SM/DOPC/cholesterol, we found that the surface pressure changes (Δπ) induced by lysenin^WT^ reached 13 mN/m, while those induced by lysenin^V88C/Y131C^ amounted to 8 mN/m (Figure 3A). Addition of 10 mM DTT increased the kinetics of surface pressure evoked by lysenin^V88C/Y131C^ to the level observed with lysenin^WT^, indicating that cysteine residue reduction allowed incorporation of lysenin^V88C/Y131C^ in a similar manner as lysenin^WT^. Negligible changes of the surface pressure were found for DOPC/cholesterol monolayers (Figure 3A). Measurement of intrinsic tryptophan fluorescence confirmed that upon binding to SM-containing liposomes under reduced conditions, lysenin^V88C/Y131C^ incorporated into the lipid membrane similarly to lysenin^WT^. In contrast, in the oxidized state, binding to liposomes left the tryptophan residues of lysenin^V88C/Y131C^ in a more hydrophilic environment (Figure 3B).

### 2.2. Effect of V88C/Y181C Mutation on Oligomerization and Pore-Forming Activity

The interaction of lysenin with an SM-containing membrane causes the lysenin molecules to self-assemble, forming highly stable oligomers [18]. Electron microscopy analysis revealed that upon binding to SM-containing liposomes, both lysenin^WT^ and lysenin^V88C/Y131C^ formed well-organized oligomeric structures distributed in a honeycomb-like pattern, regardless of whether a reducing agent was present (Figure 4A). However, in the absence of DTT, lysenin^V88C/Y131C^ clearly created less contrasting oligomeric structures. These findings may suggest that lysenin^V88C/Y131C^ and lysenin^WT^ differ in their orientation relative to the SM-containing membrane or oligomeric form [28].

Additional studies performed with lysenin^WT^ confirmed that this protein efficiently formed SDS-stable oligomers upon binding to liposomes composed of SM/DOPC/cholesterol at both 20 °C and 37 °C. The double-cysteine mutation V88C/Y131C disrupted the formation of SDS-stable oligomers, which was particularly evident when lysenin^V88C/Y131C^ was incubated with SM/DOPC/cholesterol liposomes at 20 °C. Under these conditions, the vast majority of lysenin^V88C/Y131C^ was observed in the monomer form, with only minute amounts of SDS-resistant oligomers detected (Figure 4B). However, analysis of oligomerization under native conditions indicated that, upon binding to SM-containing membranes, lysenin^V88C/Y131C^ forms oligomeric assemblies comparable to lysenin^WT^ (Appendix A). Reduction of disulfide bonds using 10 mM DTT restored the stability of lysenin^V88C/Y131C^ oligomeric complexes, which was particularly evident when SM-containing liposomes were incubated at 37 °C (Figure 4B). Further analysis of oligomer stability showed that lysenin^V88C/Y131C^ that bound to liposomes at 37 °C was more susceptible to trypsin digestion compared to lysenin^WT^ (Figure 4C). Addition of 10 mM DTT to the lysenin^V88C/Y131C^/liposome mixture increased the protection of lysenin^V88C/Y131C^ oligomers from proteolytic digestion (Figure 4C), to equal that observed with lysenin^WT^. These results suggested that oligomers of lysenin^V88C/Y131C^ formed under oxidized condition differed from lysenin^WT^ oligomers with regards to structural stability and the degree of incorporation into the lipid membrane.

The ability of lysenin to form SDS-resistant oligomers correlates with its pore-forming activity that leads to hemolysis of erythrocytes. To estimate the relative lytic activities of lysenin^V88C/Y131C^, we performed fluorometric measurements of carboxyfluorescein release from lipid vesicles and a hemolytic assay. In agreement with previous findings [17], lysenin^WT^ dose-dependently increased lipid membrane permeability, while lysenin^V88C/Y131C^ did not induce carboxyfluorescein release from liposomes (Figure 5A).

In the presence of 10 mM DTT, the pore-forming activity of lysenin^V88C/Y131C^ was restored to a level near that observed for lysenin^WT^, reaching almost 100% carboxyfluorescein release after incubation with 500 nM of protein at 37 °C (Figure 5A). Accordingly, the double-cysteine mutation also markedly decreased the hemolytic activity of lysenin for concentrations up to 500 nM, at both 20 °C and 37 °C (Figure 5B). In the presence of a reducing agent (10 mM DTT), the hemolytic activity of lysenin^V88C/Y131C^ was completely restored at both incubation temperatures (Figure 5B), reaching HA_50_ at about 25 μM and 60 μM with incubation at 20 °C and 37 °C, respectively, confirming earlier data [22]. As we expected, the presence of DTT did not change the hemolytic activity of lysenin^WT^. We also observed that the presence of reducing agent restored the lytic activity of lysenin^V88C/Y131C^ for HEK293 cells that have lower SM content in the plasma membrane than erythrocytes. Total calcein release from HEK293 cells incubated with lysenin^WT^ was observed at a concentration of 500 nM after a one-hour incubation at 37 °C, while lysenin^V88C/Y131C^ supplemented with 10 mM DTT induced ~90% of that calcein efflux (Appendix A).

### 2.3. Effects of the V88C/Y181C Mutation on Lysenin’s Structure in its Soluble Form

To gain insight into the structural changes accompanying lysenin’s interaction with the membrane, we performed HDX-MS, which can distinguish unstructured (flexible, destabilized) from highly structured (stabilized) regions of a protein based on differences in the exchange kinetics of amide protons. We applied HDX-MS to obtain information regarding the accessibility of amide protons for deuterium uptake in different regions of lysenin^WT^ and lysenin^V88C/Y131C^, both in solution and bound to the lipid membrane.

Pepsin digestion allowed to obtain high sequence coverage (over 85%) for lysenin^WT^ and lysenin^V88C/Y131C^ at different conditions, with overlapping peptides in multiple regions (Appendix A). To perform the hydrogen–deuterium exchange reaction of proteins in aqueous solution, samples were incubated with D_2_O at 20 °C for different times (10 s, 1 min, 5 min, 20 min, 2 h, or 24 h). The results obtained for 10 seconds of the hydrogen–deuterium exchange reaction revealed an intertwined pattern of protected (structured) and unprotected (unstructured) regions along the lysenin sequence (Figure 6A).

When analyzing the structural dynamics of lysenin^WT^, we found that both the N-terminal and C-terminal domains included highly dynamic regions (deuteration above 50%) and strongly stiffened regions (deuteration lower than 20%) (Figure 6A; top panel, black). The regions exhibiting the fastest exchange, with visible deuteration after 10 seconds, were detected for peptides encompassing strands β3 and β10–β11 in the N-terminal domain, and amino acid positions 180–197 and 203–209 in the C-terminal domain (Figure 6A). The hydrogen–deuterium exchange reaction for 20 min exhibited a relatively high degree of the dynamics of soluble lysenin^WT^, indicating deuteration levels of >50% along the majority of the lysenin sequence (Figure 6B; top panel, black). Few peptides exhibited strong protection against the exchange reaction, with only three regions showing <30% deuteration. The most stabilized peptides spanned amino acid positions 70–76 (strand β5) in the N-terminal domain, and two peptides in the C-terminal domain at positions 198–202 and 266–272 (Figure 6B; top panel, black). The presence of a set of relatively stable regions allowed precise organization of the flexible structure of lysenin in space.

We also performed HDX experiments for structural characterization of lysenin^V88C/Y131C^ (Figure 6A,B; top panel, blue). Based on the pattern of hydrogen–deuterium exchange, the introduction of the double-cysteine mutation and the presence of a disulfide bridge did not alter the structural dynamics of lysenin in aqueous solution. An overall view of the exchange pattern of lysenin^V88C/Y131C^ revealed no changes in the deuteration level along the entire protein sequence compared to lysenin^WT^ after 10 sec (Figure 6A; top panel, blue vs. black) and 20 min of exchange (Figure 6B; top panel, blue vs. black). This similarity was also observed for lysenin^V88C/Y131C^ in the presence of 10 mM DTT, which broke the disulfide bond between cysteine residues at positions 88 and 131 and restored lytic activity (Figure 6A,B; top panel, red vs. black). Precise analysis of differences in exchange between lysenin^WT^ and lysenin^V88C/Y131C^ (both reduced and non-reduced form) confirmed that the proteins shared similar structural stability in aqueous solution. Subtraction plots clearly indicated that the dissimilarity of deuteration between corresponding peptides of the two proteins in relation to lysenin^WT^ did not exceed 5% for hydrogen–deuterium exchange reactions of 10 sec (Figure 6C; lysenin^V88C/Y131C^ in blue; lysenin^V88C/Y131C^ with DTT in red) or 20 min (Figure 6D). Additionally, the measurement of hydrogen–deuterium exchange at different times (10 sec, 1 min, 5 min, 20 min, and 2 hours) enabled determination of the exchange kinetics in distinct peptides of lysenin^WT^ and lysenin^V88C/Y131C^. Based on this detailed analysis, we demonstrated the similar kinetics of deuterium uptake for individual regions of lysenin^WT^ and lysenin^V88C/Y131C^, both in the reduced and oxidized forms, which further confirmed the similar structural dynamics of wild-type and mutated lysenin in aqueous solution (Appendix A).

### 2.4. Changes in the Structural Dynamics of Lysenin^WT^ and Lysenin^V88C/Y131C^ in the Membrane-Bound Form

Conformational changes accompanying prepore and transmembrane pore creation may alter a protein’s solvent accessibility or hydrogen bonding network, which are both reflected by changes in the deuterium incorporation into protein. Comparison of the hydrogen–deuterium exchange patterns of lysenin^WT^ and lysenin^V88C/Y131C^ bound to SM-containing membranes enabled us to map the changes in lysenin’s conformational dynamics during pre-pore and pore formation. An overall view of the exchange patterns of lysenin^WT^ and lysenin^V88C/Y131C^ interacting with SM-containing liposomes, both with and without 10 mM DTT, indicated substantial reductions of deuterium uptake after 20 min compared to the proteins in aqueous solution (Figure 7A–C; Appendix A). The HDX patterns for lysenin^V88C/Y131C^ with 10 mM DTT and lysenin^WT^ bound to liposomes revealed that both proteins had the same structural dynamics (Figure 7D). This clearly indicated that disulfide bridge reduction in lysenin^V88C/Y131C^ reinstated the structural dynamics to the state observed for lysenin^WT^, with consequent full restoration of the lytic activity. Therefore, in further structural analyses, we used a reduced form of lysenin^V88C/Y131C^ as a model of pore-forming lysenin.

Subtraction plots provided the best visualization of the differences in exchange between corresponding peptides in lysenin^V88C/Y131C^ bound to liposomes vs. in aqueous solution, with or without DTT (Figure 7E). The subtraction plot clearly showed that lysenin^V88C/Y181C^, with or without a reducing agent, revealed decreased deuteration (stabilization) along its entire sequence upon binding to SM-containing liposomes as compared to in its soluble form. However, two regions in the N-terminal domain of lysenin^V88C/Y131C^ (both oxidized and reduced) demonstrated higher deuterium exchange (destabilization) upon binding to liposomes than its soluble form. These regions included two common peptides spanning positions 64–76 (strand β5), and one peptide spanning positions 114–118 (central part of strand β8) (Figure 7E and Figure 8D,E). We observed stronger stabilization (>60% difference between soluble and liposome-bound forms) in lysenin^V88C/Y131C^ with DTT (Figure 7E; red) compared to without DTT (Figure 7E; blue). The largest differences in deuteration level concerned two segments of the N-terminal domain of the two lysenin^V88C/Y131C^ forms.

One segment was localized in the region of amino acid positions 37–69, corresponding to strands β3 and β4, as well as the linker between strand β4 and the tongue region that is clearly less exchangeable in the reduced form of lysenin^V88C/Y131C^ (Figure 7B,E and Figure 8C,D; red). In consequence, in oxidized form of lysenin^V88C/Y131C^, the peptide encompassing the tongue region (peptide at position 52–64) showed exchange similar to that observed for the soluble form of lysenin (Figure 7A,E and Figure 8D, blue). The other segment, which showed a slightly less pronounced difference between the reduced and oxidized form of lysenin^V88C/Y131C^, included strands β10 and β11 at positions 135–153 (Figure 7B,E and Figure 8E, red vs. blue). The C-terminal domain also exhibited significant stabilization upon binding to liposomes, both with and without DTT (Figure 7E). Similar to the N-terminal domain, decrease in the hydrogen–deuterium exchange was more pronounced in the reduced form of lysenin^V88C/Y131C^ than in its unreduced form. In the reduced form, the most reduced deuteration regions (exhibiting deuteration of at least 50% lower compared to lysenin in solution) included positions 182–197, 235–248, and 282–293 (Figure 7E and Figure 8F,G,H; red).

We next compared the hydrogen–deuterium exchange pattern between liposome-bound lysenin^V88C/Y131C^ with vs. without DTT. This enabled mapping of the regions in lysenin that showed changes in structural dynamics during the prepore-to-pore transition (Figure 9; Appendix A blue vs. red). The pattern of hydrogen–deuterium exchange after 10 s of deuteration revealed very low exchange for both the non-lytic and lytic forms of lysenin. The deuteration level did not exceed 30% along the majority of the lysenin sequence, with the clearest deviation identified for the tongue (positions 52–69) in unreduced lysenin^V88C/Y131C^, which was relatively susceptible to deuteration (Appendix A; blue).

The vast majority of the both liposome-bound lysenin forms exhibited deuteration of ~20% after 10 s of exchange reaction (Appendix A). Despite the apparently similar structural dynamics, the differential plot obtained after subtraction of the exchanged fraction of liposome-bound oxidized lysenin^V88C/Y131C^ from that of reduced lysenin^V88C/Y131C^ revealed that lysenin^V88C/Y131C^ in the presence of DTT was slightly more protected to hydrogen–deuterium exchange compared with the nonlytic form (Figure 9A).

After 20 min of exchange reaction, the deuteration level was significantly increased only for lysenin^V88C/Y131C^ without DTT, highlighting the native destabilization of this protein in its prepore state (Appendix A; blue). The highest exchange (deuteration above 70%) was observed in the N-terminal fragment including positions 52–69, and in the C-terminal region including positions 235–248 and 282–293 (Figure 8D,G,H; Appendix A). We also identified fragments of both lysenin forms that showed low deuteration levels (<30% deuteration), which included peptides encompassing amino acid positions 16–27, 119–181, and 262–272 (Appendix A). These regions were strongly protected even after 2 h of deuteration (Figure 8I,J), indicating that their stabilization occurred at the prepore formation stage and was maintained until final pore formation. A differential plot provided better visualization of the changes in lysenin’s structural stability accompanying prepore-to-pore conversion (Figure 9B). Analysis revealed that lysenin generally underwent stabilization along its entire sequence during the prepore-to-pore transition, although lysenin’s stabilization in the pore structure was only slightly stronger compared with the prepore structure, with a difference not exceeding 30% of deuteration. During prepore-to-pore conversion, we observed strong decrease of deuteration level of the lysenin structure in only three regions, encompassing amino acid positions 37–69, 182–197, and 225–248 (Figure 9B). The weakest deuteration was observed for the peptide at position 52–69, corresponding to the tongue region that forms the extended β-strand that is part of the transmembrane β-hairpin (Figure 9B).

In summary, HDX-MS allowed determination of structural elements that were crucial for converting lysenin from its soluble form to the pore structure. Using this method, we identified the changes in the structural dynamics of these regions that occurred during two steps of pore formation: conversion of the soluble form of lysenin into the prepore structure, and the prepore-to-pore transition. Formation of the prepore structure was accompanied by different level of stabilization of the lysenin structure, with the exception of the β5 strand that was more destabilized in the prepore than in the soluble form (Figure 10A,B). Prepore-to-pore conversion involved progressing stabilization of the lysenin structure, but these changes were less pronounced than in the case of prepore formation, excluding the region between strands β3 and β5, and region in the C-terminal domain, encompassing strands β17–β22, that were over-stabilized in the pore structure (Figure 10B). Interestingly, destabilization of the region at amino acid positions 70–76 (β5 strand) remained unchanged during prepore-to-pore conversion, due to the fact that this region protrudes on the other side after passing through the membrane (Figure 10B).

Overall, our results demonstrated that the structural dynamics of lysenin in the prepore state significantly differed from the soluble monomer form, in both the N-terminal and C-terminal domains. The data strongly indicated that the prepore formation forced changes in lysenin’s structural dynamics that allowed conformational transition required for protein insertion into the lipid membrane and lytic pore formation.

## 3. Discussion

The interaction of lysenin with SM-containing membranes forces structural changes within the toxin molecule, leading to transmembrane pore formation. Lytic pore formation is a complex process that involves binding of lysenin’s monomeric form to the SM-containing membrane, oligomerization on the plasma membrane surface, and insertion of lysenin into the lipid membrane. In a previously proposed model, lytic pore formation is initiated by the binding of lysenin to a SM-enriched lipid membrane via the C-terminal domain of the toxin molecule. The lysenin molecule concentration at the membrane surface promotes interactions between neighboring monomers, leading to the formation of arc-shape oligomers that eventually grow into complete nonameric ring-shaped prepores [22]. It has not previously been shown that prepore formation is accompanied by structural changes in the lysenin molecule, although conformational changes during the monomer-to-prepore transition have been observed for other members of the aerolysin family, including aerolysin and epsilon toxin [29,30,31]. Finally, significant structural rearrangement within lysenin’s N-terminal domain enables insertion of the protein into the lipid membrane, and formation of a β-barrel transmembrane pore. Prior analyses of the crystal and cryo-EM structures of the lysenin pore clearly reveal the structural rearrangement that occurs during the transition of the soluble monomer structure to the membrane-inserted oligomeric state [8,22]. Conformational changes accompanying lytic pore formation lead to reorganization of a region of the N-terminal domain including three β strands (β4, β5, and β6), which consequently forces extension of the β3 and β7 strands and a single 3_10_ α-helix, promoting their assembly into the twisted β-hairpin. This rearrangement results in downward rotation of the N-terminal domain and a bend in the middle of the structurally intact region of the N-terminal domain (collar), allowing insertion of the twisted β-hairpins of adjacent monomers into the membrane and formation of the β-barrel. Although analysis of the lysenin pore structure has provided new insights into the final membrane-inserted state, structural characterization of the intermediate steps along the pore-formation pathway is required to derive the complete molecular mechanism.

In our present study, we endeavored to gain information about the dynamics of conformational changes occurring during prepore and pore formation. To this end, we characterized the interaction of the V88C/Y131C lysenin mutant with lipid membranes and determined its structural dynamics in both the water-soluble and membrane-bound forms. This double mutation enables the creation of a disulfide bridge between cysteine residues at positions 88 and 131, which prevents insertion loop displacement and consequently abolishes pore formation. On the other hand, this disulfide bond does not affect lysenin’s ability to form nonameric oligomers. Reduction of the disulfide bond leads to disengagement and unfolding of the insertion loop, which completely restores pore-forming activity to the level observed for lysenin^WT^. As previously shown for another pore-forming protein, perfringolysin O, the introduction of single point mutations induces local changes in the protein structure and may also cause global changes in the protein structure without influencing the secondary structure [25,32]. It has also been shown that the introduction of a point mutation outside of the area involved in membrane binding can alter the interaction between the receptor-binding motif (domain) and the lipid bilayer, indicating both structural and functional coupling of distant and spatially separated domains of PFO [33]. These prior findings indicated a need to verify the binding properties of lysenin^V88C/Y131C^. Extensive analysis of lysenin’s interaction with the lipid bilayer revealed that the double-cysteine mutation and the creation of a cysteine bridge between strands β6 and β10 did not affect lysenin’s recognition and initial binding to SM or oligomer formation. However, subtle differences in the stability of this interaction were detectable depending on the redox state of the V88C and Y131C residues. The decreased surface pressure of the lipid monolayer in the presence of the oxidized form of lysenin^V88C/Y131C^, the increased dissociation of this protein upon binding to SM-containing liposomes, less contrasting oligomers in ultrastructural analysis, and higher susceptibility to proteolysis indicated weaker anchoring of lysenin oligomers in the lipid membrane, thus confirming impairment of lysenin’s insertion into the lipid membrane in the prepore state. Based on these properties of lysenin^V88C/Y131C^, we decided to use the soluble form of this protein as a model for studying the starting structure during pore formation, and to use the liposome-bound lysenin^V88C/Y131C^ under different redox conditions as model to study the prepore and pore forms.

Previous data have revealed that the flexibility of two regions: the random coil region (V157–E167) connecting the C-terminal domain to the structurally intact region of the N-terminal domain (collar) and the tongue region (M44–G67) play crucial roles in the transition from the soluble monomer to the membrane-inserted state [22]. The high dynamics of the linker between the N-terminal and C-terminal domains allows lysenin to collapse toward the lipid membrane and the flexibility of the tongue enables formation of the β-hairpin that incorporates into the lipid bilayer. Our present HDX results confirmed that both regions were highly dynamic in the soluble form of lysenin. However, both were strongly stabilized upon incorporation into the lipid bilayer and pore formation (Figure 10). Interestingly, the linker between the N-terminal and C-terminal domains was strongly stabilized already in the prepore state, and the stabilization was similar as in the pore. In contrast, the tongue region showed a similar (region between strands β4 and β5) or slightly reduced (region between strands β3 and β4) flexibility in the prepore state compared to the soluble monomer of lysenin. We also demonstrated that prepore formation was accompanied by stabilization of β strands embedded in the collar region, and exposure of the hydrophobic β5 strand to the solvent, which ultimately created the transmembrane β-hairpin and inserted it into the lipid bilayer. These findings indicated that the gradual decrease in flexibility of the tongue fragment, and the stability of the β-strand network in the collar region, play crucial roles in the prepore-to-pore transition.

In the previously described prepore model, the N-terminal twisted β-sheet remains structurally intact relative to lysenin in its soluble state. Therefore, the β2 strand is faced toward the insertion loop of the neighboring lysenin molecule and interacts and arranges itself parallel to the β10 strand of the adjacent monomer only upon unwinding of the tongue and tilting by 45° from the collar [22]. The network of interactions between amino acid residues maintains the correct dynamics and conformation of lysenin that are required for the structural rearrangement involved in lytic pore formation.

Our present findings support the possibility that subtle intramolecular rearrangements in lysenin occur at prepore formation step, which ultimately facilitate membrane insertion and pore formation. Despite the salt bridges are not dominant factors that governs lysenin’s stability in the soluble monomeric form, they are one type of interaction crucial for structural rearrangement of monomer during pore formation (Figure 11A–E) [8].

The formation of intermolecular salt bridges between adjacent monomers during oligomerization is the mechanism controlling the transition of prepore to pore in perfringolysin O and may commonly occur among pore-forming proteins [34].

Electrostatic interactions are also important for maintaining contact between adjacent monomers and the stability of the β-barrel pore (Figure 11F) [22]. Our results suggested that stiffening of the N-terminal twisted β-sheet formed by β2, β8, β9, β10, and β11 may force the breaking of salt bridges between the α-helix and tongue (R52–E92), β8 strand and tongue (H58–D121), and β5 and β10 strands (H81–E135), thus leading to slight stabilization of the insertion loop and consequently exposing the β5 strand to solvent and increasing its conformational dynamics (Figure 8A and Figure 11). This orientation is energetically unfavorable; therefore, to decrease the energy state, the insertion loop must be displaced, thus allowing unfolding of the β-hairpin and formation of the β-barrel pore structure.

Based on our present findings and previous work, we also conclude that the interaction of lysenin with an SM-containing membrane in the prepore state moderately affect the structural dynamics of the C-terminal domain, including the regions that interact with the phosphocholine head group of SM, including peptide 180–192 (K185), peptide 225–233 (S227, Q229, Y233), and peptide 282–293 (Y282) [9]. Surprisingly, the prepore-to-pore transition was accompanied by strong stabilization of the C-terminal part of lysenin, even at regions that are not critical for SM binding. This may be explained by the structural rearrangement within the N-terminal domain of lysenin during pore formation, which forces the proper orientation and stability of the lysenin pore in the lipid membrane, including stabilization of the C-terminal domain of lysenin. This observation indicates that the receptor-binding domain’s role is not limited only to the recognition of targets on cell membranes but is also crucial for stabilizing the oligomeric complex of the lysenin in the prepore form and the formation of functional pore structures. It can be assumed that initial electrostatic interaction between C-terminal domain of lysenin and POC forces changes in the structural dynamics of lysenin, which lead to stabilization of structural elements in the N-terminal domain, including the region responsible for the interaction of lysenin with the SM acyl chain tail (β2 strand containing Y24 and Y26). Stabilization of this region was found already at the prepore state and is maintained also in the pore form. Similar effect is observed for β10 strand that interacts with β2 of adjacent monomer during pore formation upon tilting from the collar and unwinding of the tongue [22]. Stabilization of these two elements at the prepore stage may indicate that initial interaction of lysenin with the lipid membrane induces dynamic changes first in the regions responsible for assembly of oligomeric complex, and only then in the regions that participate in the insertion of lysenin into the membrane. Thus, we suggest that the initial interaction of β2 and β10 of adjacent monomers takes place even before unwinding of the tongue. However, despite the fact that a lot of detailed information on the structural dynamics of lysenin at various stages of pore formation has been obtained, extensive research is still required to fully understand the mechanism of pore formation.

The above-presented results demonstrate that HDX-MS is an excellent technique that provides valuable data regarding the structural dynamics of proteins at different steps of their interactions with lipid membranes. Our results obtained using this method show that binding to SM-containing membranes caused changes in the structural dynamics of lysenin, as early as in the prepore state. 

At this stage, interaction between adjacent monomers induced conformational stabilization of lysenin, which was initiated at the membrane-binding site in the C-terminal domain of lysenin and was transmitted via the N-terminal twisted β-sheet to the region forming the transmembrane domain of the pore (Figure 12). Structural stabilization of these regions forced a structural rearrangement within the PFM, which was accompanied by stabilization of insertion loop and was required for transmembrane lytic pore formation.

Overall, our present results demonstrated that the HDX-MS technique perfectly complements structural studies based on crystallography and high-resolution microscopy, bringing us closer to a full understanding of the molecular mechanism of pore formation by lysenin.

## 4. Materials and Methods

### 4.1. Expression and Purification of Recombinant Proteins

The plasmids expressing lysenin^WT^ and lysenin^V88C/Y131C^ with a HIS tag at the N terminus were generated from the pGEX4T-lysenin WT template, constructed as previously described [17]. The V88C/Y131C substitution in the lysenin sequence was generated by site-directed mutagenesis using two primer pairs: 5′-CATGAAGAATCCCAATGTAGCATGACGGAAAC-3′ and 5′-GTTTCCGTCATGCTACATTGGGATTCTTCATG-3′ to introduce the V88C mutation, and 5′-GATATTGAATACATGTGTTTGATTGATGAAGTC-3′ and 5′-GACTTCATCAATCAAACACATGTATTCAATATC-3′ for the Y131C substitution. The cDNA fragments encoding the two proteins were amplified by PCR, and subcloned into the pET28a(+) vector (Novagen, Madison, USA) using BamHI/HindIII restriction sites. The recombinant vectors pET28a-lyseninWT and pET28-lyseninV88C/Y131C were transformed into the *E. coli* BL21(DE3) and SHuffle T7 Express strains, respectively. The produced proteins (Appendix A) were purified on a Ni Sepharose 6 Fast Flow column (GE Healthcare, Munich, Germany) and dialyzed against 30 mM Tris buffer (pH 7.5) containing 150 mM NaCl. Finally, the collected samples were mixed with 10% (*v*/*v*) glycerol, frozen in liquid nitrogen, and stored at −80 °C.

### 4.2. Protein-Lipid Overlay Assay

To test the specificity of lipid recognition by lysenin^V88C/Y131C^, we spotted nitrocellulose membranes with 1 µL of lipid samples containing various amounts (5–50 pmol) of 1,2-dioleoyl-sn-glycero-3-phosphocholine (DOPC; Avanti), 1,2-dipalmitoyl-sn-glycero-3-phosphocholine (DPPC; Avanti), cholesterol (Sigma-Aldrich, Munich, Germany), or bovine brain SM (Sigma-Aldrich, Munich, Germany) in a chloroform:methanol:water mixture (1:1:0.3, *v*/*v*). The membranes were blocked for 1 h at 20 °C with 1% gelatin and 1% polyvinylpyrrolidone in Tris-buffered saline containing 0.03% Tween-20 (TBST) and then incubated for 2 h with 100 nM lysenin^WT^ or lysenin^V88C/Y131C^ and processed as previously described [14].

### 4.3. Surface Plasmon Resonance

We examined the interaction of lysenin^WT^ and lysenin^V88C/Y131C^ with large unilamellar vesicles (LUVs) using a BIAcore 3000 instrument (GE Healthcare, Munich, Germany) equipped with an L1 chip. LUVs composed of SM/DOPC/cholesterol (molar ratio 1:2:1) were prepared as previously described, yielding a final lipid concentration of 1 mM [17]. The liposomes were deposited onto the L1 chip surface, at a flow rate of 1 μL/min for 10 min, using an amount that yielded 7000 resonance units (RU). Measurements were conducted using 1 μM protein at a flow rate of 5 μL/min for 300 s. Then, the dissociation of the deposited protein was followed for an additional 300 s.

### 4.4. Surface Pressure Measurement

These experiments were performed using a NIMA Technologyn tensiometer model PS3 (Coventry, UK) at 20 ± 1 °C in darkness under an argon atmosphere, as previously described [13]. The water sub-phase was buffered with 30 mM Tris-HCl (pH 7.5). To measure the surface pressure of mixed lipid monomolecular layers, a lipid monolayer comprising SM/DOPC/cholesterol (molar ratio 1:2:1) or DOPC/cholesterol (molar ratio 2:1) was deposited at the argon–water interface from a chloroform solution. A 40 μM final concentration of lysenin^WT^ or lysenin^V88C/Y131C^ (with or without 10 mM DTT) was injected into the 12 mL of buffer sub-phase.

### 4.5. Oligomerization Assay

To prepare small unilamellar vesicles (SUVs) composed of SM/DOPC/cholesterol (1:2:1 molar ratio, 1 mM total phospholipid concentration), we resuspended the lipid film in buffer containing 30 mM Tris and 150 mM NaCl (pH 7.5). This liposome suspension was vortexed for 5 min and then subjected to six freeze-thaw cycles, and sonication (30 min, 0.3 cycle, amplitude 33%). After centrifugation at 10,000× *g* (30 min, 4 °C), the pelleted SUVs were resuspended in buffer containing 30 mM Tris and 150 mM NaCl (pH 7.5), and incubated with 1 μM lysenin^WT^ or lysenin^V88C/Y131C^ (with or without 10 mM DTT) for 30 min at 25 °C and 37 °C. The samples were then pelleted by centrifugation at 10,000× *g* (30 min), suspended in 20 μL of SDS loading buffer (40% glycerol (*v*/*v*), 25% SDS (*w*/*v*), and 0.1% bromophenol blue (*w*/*v*)), and analyzed by SDS-PAGE.

The oligomeric complexes of lysenin were then analyzed for their susceptibility to proteolysis. The samples of proteins bound to liposomes were suspended in buffer containing 30 mM Tris and 150 mM NaCl (pH 7.5) and supplemented with trypsin (trypsin:protein ratio 1:20, *w*/*w*; Promega). After incubation for 2 hours at 37 °C, the samples were diluted with SDS loading buffer and analyzed by SDS-PAGE.

### 4.6. Measurement of Intrinsic Protein Fluorescence

To determine the intrinsic protein fluorescence, we measured the tryptophan emission spectra of lysenin^WT^ and lysenin^V88C/Y131C^, both in solution and bound to liposomes. SUV preparation and lysenin binding to lipid vesicles were performed as described above. Tryptophan emission was excited at 280 nm, and fluorescence emission was recorded from 300 to 400 nm at 1 nm/s. All measurements were made using a JASCO FP 6500 spectrofluorimeter at 25 °C. The spectral band widths were 3 nm for both excitation and emission.

### 4.7. Electron Microscopy

Ultrastructural studies of lysenin oligomeric complexes were performed as described by Kwiatkowska et al. [14], with some modifications. To prepare multilamellar lipid vesicles (MLVs), the lipid film (composed of SM/DOPC/cholesterol in a 1:2:1 molar ratio) was suspended in buffer containing 30 mM Tris and 150 mM NaCl (pH 7.5), vortexed, and sonicated for 2 min at 4 °C (30 min, 0.3 cycle, amplitude 33%). After centrifugation at 15,000× g (15 min, 4 °C), the liposomes were resuspended in buffer containing 30 mM Tris and 150 mM NaCl (pH 7.5) and incubated for 30 min at 20 °C with 7 µM lysenin^WT^ or lysenin^V88C/Y131C^, with or without 10 mM DTT. After incubation, the liposomes were fixed with 1% glutaraldehyde in PBS for 20 min at 20 °C and transferred onto poly-L-lysine-coated and formvar/carbon-treated grids for 20 min. Finally, the samples were washed twice with PBS and once with water, counterstained with 2% uranyl acetate, and examined under a JEM-1200EX (JEOL) microscope.

### 4.8. Pore-Forming Activity

SUVs composed of SM/DOPC/cholesterol (1:2:1 molar ratio) were prepared as described above, in the presence of 50 µM 6-carboxyfluorescein (Sigma, Munich, Germany). These SUVs were pelleted, resuspended in buffer containing 30 mM Tris and 150 mM NaCl (pH 7.5), and incubated with 1 μM lysenin^WT^ or lysenin^V88C/Y131C^ with or without 10 mM DTT. The suspension was incubated for 30 min at 20 °C, and then centrifuged. The level of released carboxyfluorescein in the supernatant was measured using a Jasco FP 6500 spectrofluorometer at an excitation/emission of 492/517 nm. The results were expressed as the percentage of maximum carboxyfluorescein release from SUVs, as triggered by addition of 0.1% Triton X-100.

### 4.9. Hemolytic Activity

The hemolytic activity of lysenin^WT^ and lysenin^V88C/Y131C^ was measured as previously described [14]. Briefly, serial protein dilutions (up to 1 µM) were incubated for 45 min at 20 °C and 37 °C, with 7 × 10^7^ sheep red blood cells (RBCs), with or without 10 mM DTT. Then the samples were centrifuged at 200× *g* (5 min, 4 °C), and the level of hemoglobin released from RBCs was estimated by measurement of the supernatant’s absorbance at 405 nm using a NanoDrop 2000c Spectrophotometer (Thermo Fisher Scientific, Waltham, MA, USA). The results were expressed as the percentage of maximal hemolysis (100%), obtained by osmotic lysis of RBCs with water.

### 4.10. Hydrogen–Deuterium Exchange Measurements

Samples for hydrogen–deuterium exchange were prepared following a previously described procedure, with some modifications [24,25]. Briefly, lipid-bound samples were prepared by incubation of 4 µM lysenin^WT^ and lysenin^V88C/Y131C^ (with or without 10 mM DTT) with 2 mM SUVs composed of SM/DOPC/cholesterol (molar ratio 1:2:1) at room temperature for 30 min. Then, the samples were centrifuged at 10,000× *g* (10 min, 4 °C), and the pellets were resuspended in 30 mM Tris with 150 mM NaCl (pH 7.5) supplemented with 2% n-dodecyl-β-D-maltoside (DDM) and incubated at room temperature for 1 h with shaking. Next, the samples were again centrifuged at 10,000× *g* (10 min, 4 °C), and the supernatant containing extracted oligomers was subjected to HDX-MS.

The HDX-MS measurements were compared between lysenin^V88C/Y131C^ and lysenin^WT^, without liposome binding (aqueous solution) and upon binding to liposomes (extracted oligomers). The hydrogen–deuterium exchange reaction was initiated by adding 5 µL protein sample to 45 µL D_2_O Reaction buffer (30 mM Tris and 150 mM NaCl, pH 7.5). The reaction was carried out for the required time period (10 s, 1 min, 5 min, 20 min, 120 min, or 24 h) and was quenched by adding the reaction mixture to 10 μL pre-chilled D_2_O stopping buffer (2 M glycine and 150 mM NaCl, pH 2.4). Finally, the samples were injected onto an immobilized pepsin column (Poroszyme; ABI), and the obtained peptides were further separated using the nanoACQUITY ultra-performance liquid chromatography (UPLC) system, followed by mass measurements on the SYNAPT G2 HDMS mass spectrometer (Waters, Milford, MA, USA). Peptide identification was based on a list of peptic peptides obtained for a non-deuterated sample using ProteinLynx Global Server software (Waters, Milford, MA, USA), as previously described [25]. HDX data analysis was performed using the DynamX 2.0 program (Waters, Milford, MA, USA). All measurements were repeated in triplicate. All controls were performed, including a back-exchange control and a carry-over effect control.

## Figures and Tables

**Figure 1 toxins-11-00462-f001:**
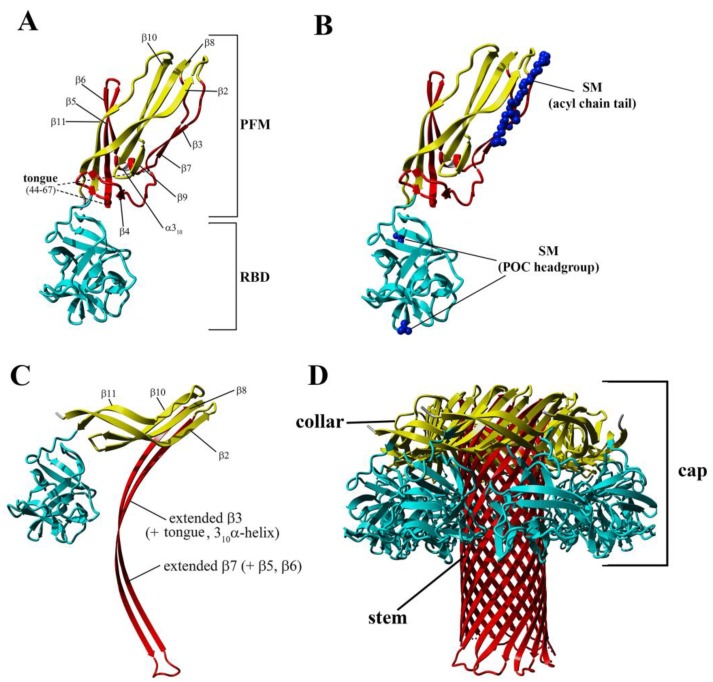
Crystal structure and structural rearrangement of the monomeric form of lysenin upon membrane binding. (**A**–**C**) Overall crystallographic structures of the water-soluble monomer of lysenin (**A**) and monomeric lysenin bound to sphingomyelin (SM) (**B**) (PDB ID: 3ZXG) as obtained by de Colibus [9], and a protomer from the lysenin pore (PDB ID: 5EC5) [22] (**C**). (**D**) Mushroom-like structure of a lysenin pore. Turquoise indicates the C-terminal domain (RBD, receptor-binding domain); yellow indicates the structurally intact region of the N-terminal domain; red indicates the insertion loop of the N-terminal domain. SM is represented as ball model and colored in blue.

**Figure 2 toxins-11-00462-f002:**
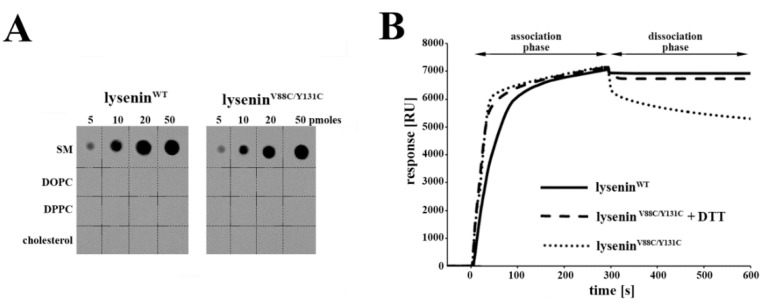
Binding of lysenin to sphingomyelin-containing liposomes. (**A**) Lipid-overlay assay revealed selective recognition of SM by recombinant lysenin^WT^ and lysenin^V88C/Y131C^. (**B**) Surface Plasmon Resonance (SPR) sensorgrams of the binding of lysenin^WT^ (

), lysenin^V88C/Y131C^ (●●●), and lysenin^V88C/Y131C^ with 10 mM 1,4-Dithiothreitol (DTT) (**▬ ▬**) to SM-containing liposomes (composed of SM/1,2-dioleoyl-sn-glycero-3-phosphocholine (DOPC)/cholesterol) that were immobilized on the surface of an L1 sensor chip. Sensorgrams were performed in triplicate, and one representative experiment is shown.

**Figure 3 toxins-11-00462-f003:**
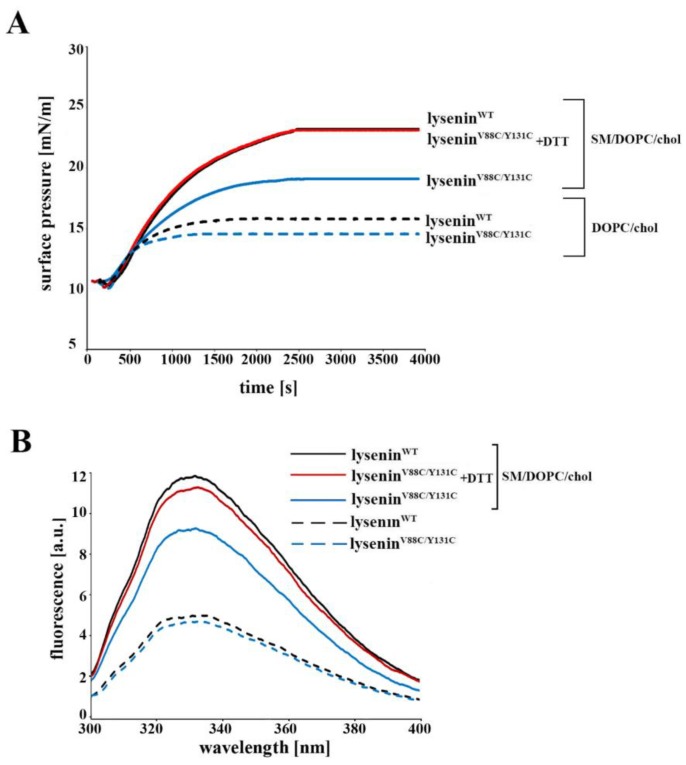
Effect of V88C/Y131C mutation on lysenin’s interaction with SM-containing layers. (**A**) Surface pressure changes caused by penetration of lysenin^WT^ (solid, black line), lysenin^V88C/Y131C^ with 10 mM DTT (solid, red line), or lysenin^V88C/Y131C^ without 10 mM DTT (solid, blue line) into a lipid monolayer containing SM/DOPC/cholesterol or DOPC/cholesterol (dashed, black line; lysenin^WT^), (dashed, blue line; lysenin^V88C/Y131C^). Plots represent one of three experiments. (**B**) Tryptophan emission spectra of lysenin^WT^ (solid, black line), lysenin^V88C/Y131C^ with 10 mM DTT (solid, red line), or lysenin^V88C/Y131C^ without 10 mM DTT (solid, blue line) bound to liposomes composed of SM/DOPC/cholesterol or in aqueous solution (dashed, black line; lysenin^WT^) (dashed, blue line; lysenin^V88C/Y131C^). Plots are representative of three independent measurements.

**Figure 4 toxins-11-00462-f004:**
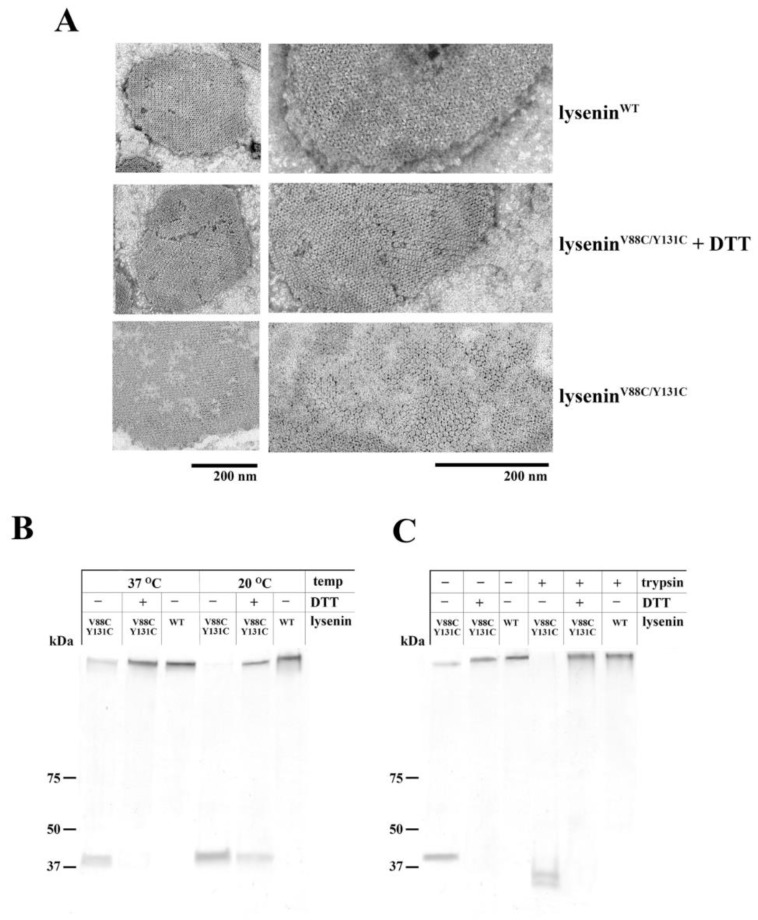
Oligomerization of lysenin^WT^ and lysenin^V88C/Y131C^ upon binding to SM-containing liposomes. (**A**) Electron microscopy images of SM-containing membranes with bound oligomers of lysenin^WT^ and lysenin^V88C/Y131C^ with or without 10 mM DTT. Oligomers were formed after incubation of 7 μM lysenin with MLVs composed of SM/DOPC/cholesterol for 30 min at 20 °C. Samples of oligomers bound to liposomes were transferred onto electron microscopy grids, and negatively stained. Scale bars, 200 nm. (**B**) SUVs composed of SM/DOPC/cholesterol were incubated with 1 μM lysenin^WT^ or lysenin^V88C/Y131C^ (with or without 10 mM DTT) for 30 min at 20 °C or 37 °C. Pelleted liposomes were subjected to SDS-PAGE (4–20% gradient) and analyzed for the presence of lysenin monomers and oligomers. Molecular weight standards are shown on the left. (**C**) Protection of lysenin against trypsin digestion after binding to liposomes. Liposomes were incubated for 30 min at 37 °C with lysenin^WT^ or lysenin^V88C/Y131C^ (with or without 10 mM DTT), and then treated with trypsin (marked as +) or not (marked as −). Protease digestion products were analyzed by 4–20% gradient SDS-PAGE with Coomassie Brilliant Blue staining.

**Figure 5 toxins-11-00462-f005:**
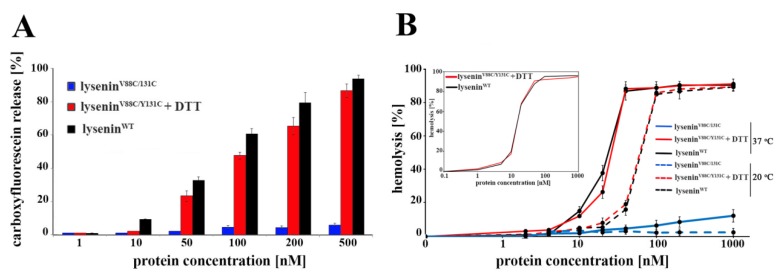
Effects of V88C/Y131C mutation on lysenin pore formation and hemolytic activity. (**A**) Lysenin’s pore-forming activity was determined by fluorometric measurement of carboxyfluorescein release from SUVs composed of SM/DOPC/cholesterol after a 30-min incubation at 37 °C with different concentrations of lysenin^WT^ (black bars), lysenin^V88C/Y131C^ (blue bars) and lysenin^V88C/Y131C^ with 10 mM DTT (red bars). The results were expressed as a percentage of the total carboxyfluorescein released from SUVs following treatment with 0.1% Triton X-100. (**B**) Hemolytic activity of lysenin^WT^ (black line), lysenin^V88C/Y131C^ (blue line) and lysenin^V88C/Y131C^ with 10 mM DTT (red line). RBCs were incubated for 45 min with indicated concentrations of the recombinant proteins at 37 °C (soiled lines) and 20 °C (dashed lines). The amount of released hemoglobin was expressed as a percentage of the total amount of hemoglobin released by osmotic lysis of RBCs. The results represent the mean ± SD of three experiments.

**Figure 6 toxins-11-00462-f006:**
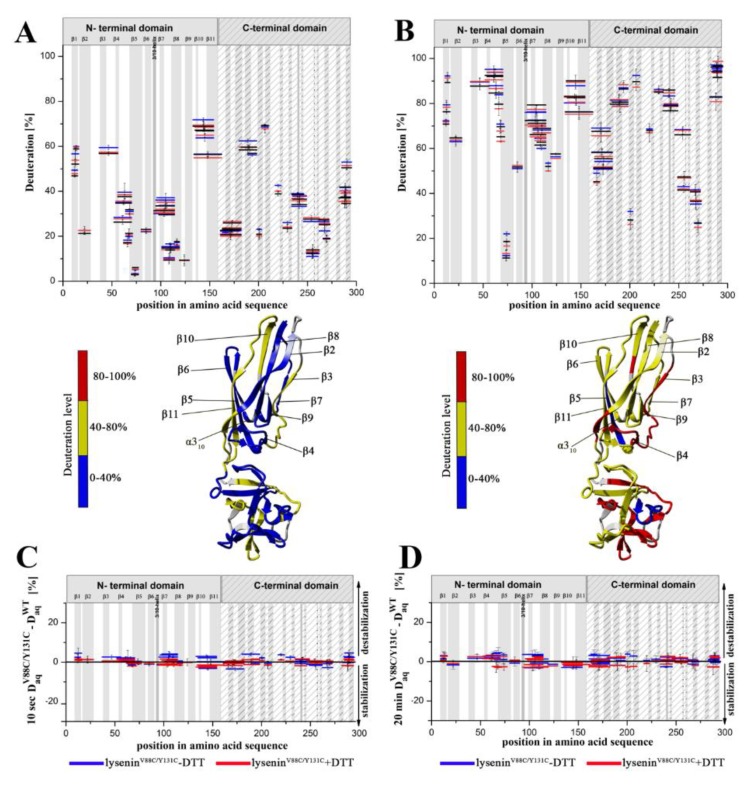
Hydrogen–deuterium exchange patterns for lysenin^WT^ and lysenin^V88C/Y131C^ in aqueous solution. (**A**,**B**) Plots show the hydrogen–deuterium exchange pattern of amide protons (y axis) for lysenin^WT^ (black), lysenin^V88C/Y131C^ without DTT (blue), and lysenin^V88C/Y131C^ with 10 mM DTT (red) in aqueous solution (top panels). Samples were incubated with deuterium buffer for 10 s (**A**) or 20 min (**B**) Peptides are represented with horizontal bars indicating their lengths and positions in the lysenin amino acid sequence (horizontal axis). Regions organized into β strands and the 3/10 helix of the N-terminal and C-terminal domains are indicated at the top of the graph. Y-axis error bars indicate standard deviations calculated from at least three independent experiments. The HDX data are represented with color-coding on schematic representations of the monomeric crystallographic structure of lysenin (bottom panels). Color coding: red regions: 80–100% deuteration; yellow regions: 40–80% deuteration; blue regions: 0–40% deuteration; grey regions: fragments not covered by the data. (**C**,**D**) Subtraction plots of the HDX patterns were generated to better visualize changes in the exchange patterns of lysenin^V88C/Y131C^ vs. lysenin^WT^ in aqueous solution. Data were obtained by subtracting the fraction of hydrogen–deuterium exchange of lysenin^WT^ from that of lysenin^V88C/Y131C^ without DTT (blue) or lysenin^V88C/Y131C^ with 10 mM DTT (red) after 10 s (**C**) or 20 min (**D**) deuteration in aqueous solution. Positive and negative values, respectively, indicate regions that were destabilized and stabilized in the lysenin mutant. Error bars were calculated as the square root of the sum of variances of the subtracted numbers.

**Figure 7 toxins-11-00462-f007:**
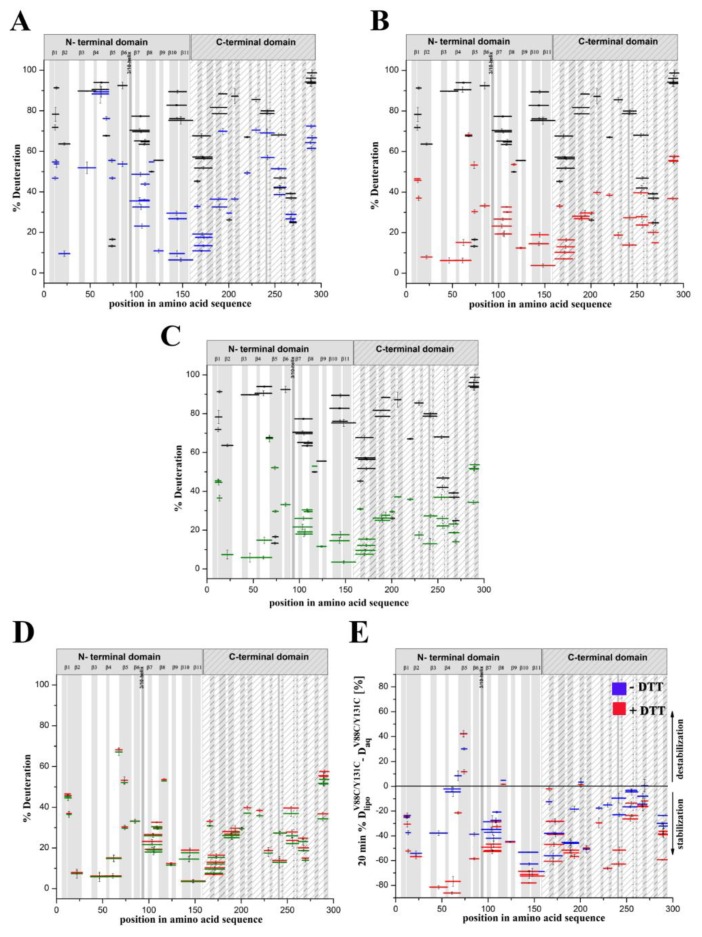
Comparison of hydrogen–deuterium exchange patterns for lysenin^V88C/Y131C^ in aqueous solution versus after binding to liposomes. (**A**–**C**) Plots show the hydrogen–deuterium exchange patterns of amide protons (vertical axis) for lysenin^V88C/Y131C^ bound to SM-containing liposomes (**A**; blue), lysenin^V88C/Y131C^ bound to SM-containing liposomes with 10 mM DTT (**B**; red), and lysenin^WT^ (**C**; green) compared to their soluble forms in aqueous solution (**A**–**C**; black) after 20 min of exchange. (**D**) Hydrogen–deuterium exchange patterns of amide protons of lysenin^V88C/Y131C^ bound to SM-containing liposomes with 10 mM DTT (red) compared to lysenin^WT^ bound to SM-containing liposomes (green). (**E**) Subtraction plot showing the difference in hydrogen–deuterium exchange patterns between lysenin^V88C/Y131C^ in aqueous solution versus bound to liposomes. Data were obtained by subtracting the fraction of hydrogen–deuterium exchange of lysenin^V88C/Y181C^ in aqueous solution from that of liposome-bound lysenin^V88C/Y181C^ with 10 mM DTT (red) or without DTT (blue) after 20 min of exchange reaction. Positive and negative values, respectively, indicate regions that were destabilized and stabilized in lysenin^V88C/Y181C^ upon binding to liposomes. Error bars were calculated as the square root of the sum of variances of the subtracted numbers. The peptides’ positions in the protein sequence and lengths are shown on the x-axis. The y-axis error bars show standard deviations calculated from at least three independent experiments.

**Figure 8 toxins-11-00462-f008:**
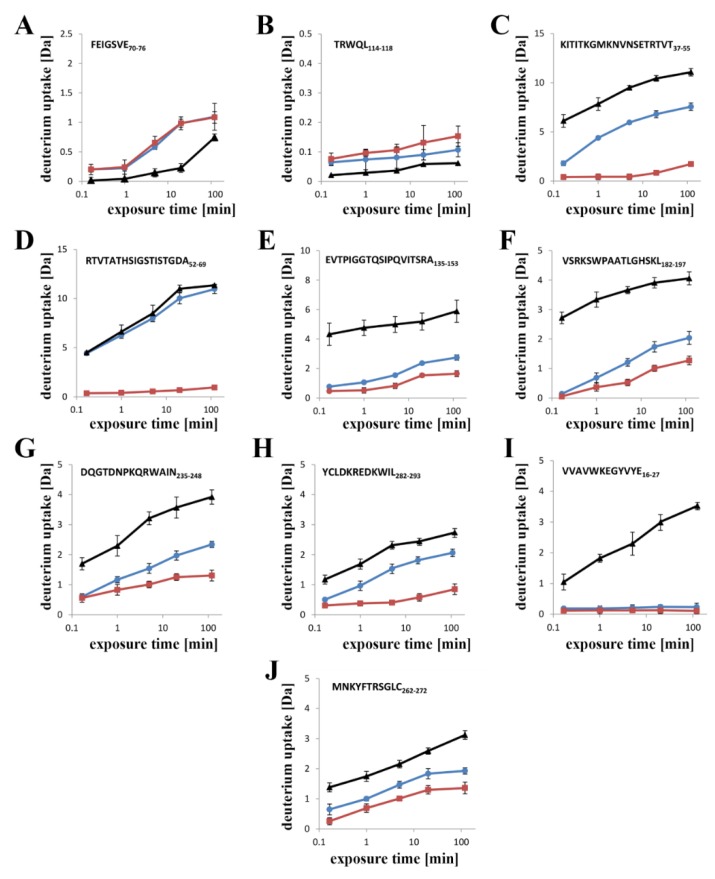
Kinetics of hydrogen–deuterium exchange in selected lysenin peptides. Deuteration was measured after exposure of lysenin^V88C/Y131C^ in solution (black), and bound to liposomes with DTT (red), bound to liposomes without DTT (blue) to D_2_O for 10 s, 1 min, 5 min, 20 min, 60 min, or 120 min and is shown on a linear time scale. Plots show the patterns of hydrogen-deuterium exchange for lysenin peptide 70–76 FEIGSVE (**A**); peptide 114–118 TRWQL (**B**); peptide 37–55 KITITKGMKNVNSETRTVT (**C**); peptide 52–69 RTVTATHSIGSTISTGDA (**D**); peptide 135–153 EVTPIGGTQSIPQVITSRA (**E**); peptide 182–197 VSRKSWPAATLGHSKL (**F**); peptide 235–248 DQGTDNPKQRWAIN (**G**); peptide 282–293 YCLDKREDKWIL (**H**); peptide 16–27 VVAVWKEGYVYE (**I**); and peptide 262–272 MNKYFTRSGLC (**J**). Each curve represents the average of three independent exchange experiments. Error bars represent S.D.

**Figure 9 toxins-11-00462-f009:**
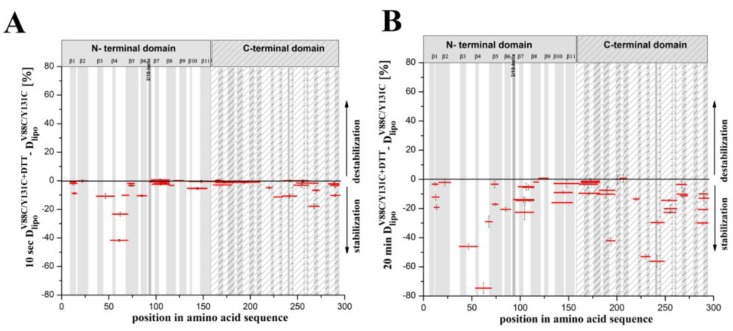
Time-dependent hydrogen–deuterium exchange patterns of lysenin^V88C/Y131C^ upon binding to liposomes. (**A**,**B**) Changes in the hydrogen–deuterium exchange pattern of proteins upon binding to liposomes for 10 sec (**A**) and 20 min (**B**) of exchange. Data were obtained by subtracting the fraction of hydrogen–deuterium exchange of liposome-bound lysenin^V88C/Y131C^ from lysenin^V88C/Y131C^ supplemented with 10 mM DTT after 10 s (**A**) or 20 min (**B**) of deuteration (red). Positive and negative values on the axis, respectively, indicate regions that were destabilized and stabilized in lysenin^V88C/Y131C^ supplemented with 10 mM DTT upon binding to liposomes. Error bars were calculated as the square root of the sum of variances of the subtracted numbers.

**Figure 10 toxins-11-00462-f010:**
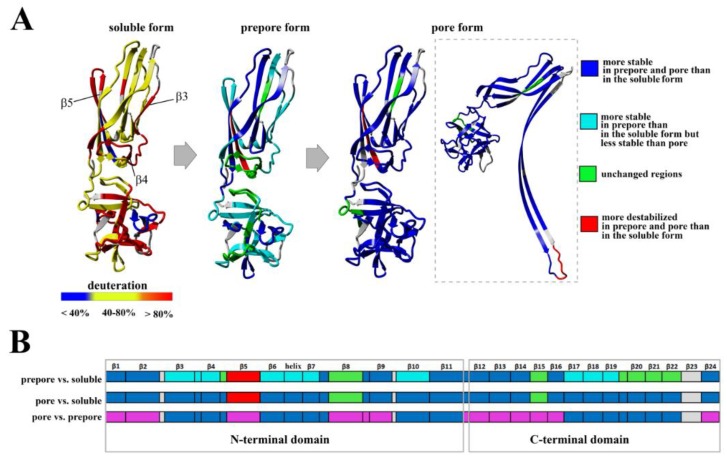
Simplified presentation of structural elements of lysenin that were stabilized and destabilized at different steps of pore formation. (**A**,**B**) HDX-MS results for 20 min of deuterium exchange reaction of soluble and liposome-bound lysenin^V88C/Y131C^ with or without DTT, overlaid on the lysenin structure (PDB:3ZXG) (**A**), and along the lysenin sequence (**B**). (**A**) In the water-soluble form structure, red indicates deuterium exchange in the range of 80–100%, yellow indicates 40–80%, and blue indicates <40%. In the prepore form structure, turquoise indicates peptides that were more stable in liposome-bound lysenin^V88C/Y131C^ (prepore form) than in its soluble form, but less stable than in liposome-bound lysenin^V88C/Y131C^ supplemented with DTT (pore); blue indicates peptides that were more stable in both the prepore and pore forms than in the soluble form; green indicates regions that were unchanged in the prepore form compared to the soluble form; red indicates peptides that were more destabilized in both the prepore and pore forms than in the soluble form; and grey indicates regions not covered by the data. In the pore form structure, blue represents structural elements that were more stable in the pore than in the soluble form of lysenin; green indicates regions that were unchanged in the pore form compared to the soluble form; and red indicates structural elements that were more destabilized in the pore form than in soluble lysenin. The square illustrates the results overlaid on the protomer of the lysenin pore structure (PDB: 5EC5). (**B**) Differences in the stabilization and destabilization of structural elements between the soluble and prepore forms (upper diagram) and between the soluble and pore forms (middle diagram). Color legend is the same as in (**A** prepore form). The lower diagram shows differences in the stabilization and destabilization between the prepore and pore forms. Blue indicates structural elements that were more stable in the pore than in the prepore form; and magenta indicates structural elements that were unchanged during the prepore-to-pore transition.

**Figure 11 toxins-11-00462-f011:**
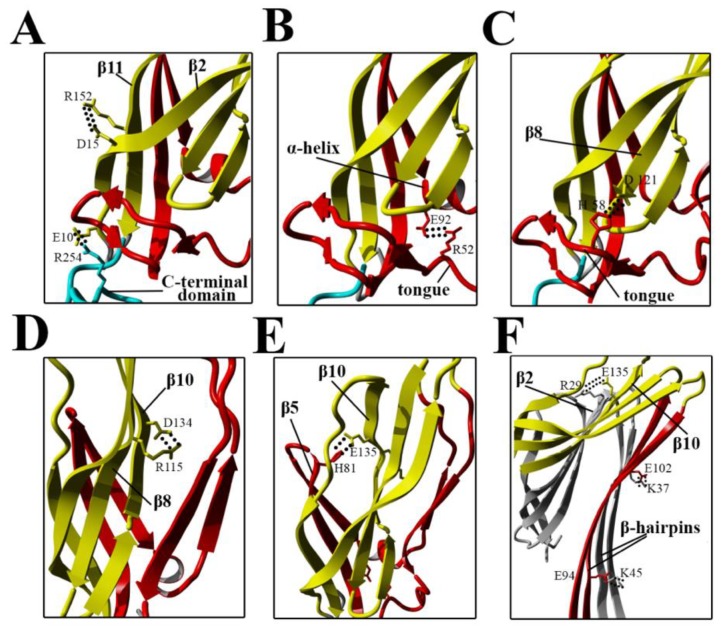
Locations of amino acid residues involved in salt bridge formation in the N-terminal domain of lysenin. Residues involved in salt bridge formation are shown as sticks. Salt bridges (dotted lines) in the soluble form of lysenin are shown between E10–R254 and D15–R152 (**A**), R52–E92 (**B**), H58–D121 (**C**), R115–D134 (**D**), and H81–E135 (**E**). (**F**) Salt bridges in the lysenin pore are shown between R29–E135, K45–E94, and K37–E102 of adjacent monomers. Secondary structures are colored as in Figure 1. Turquoise indicates the C-terminal domain (RBD, receptor-binding domain); yellow indicates the structurally intact region of the N-terminal twisted β-sheet (collar); and red indicates the N-terminal domain.

**Figure 12 toxins-11-00462-f012:**
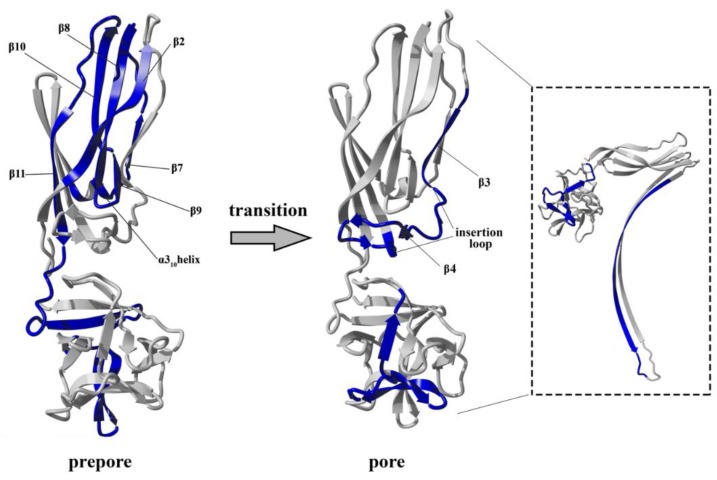
Schematic representation of the differences in structural stability between the soluble, prepore and pore forms of lysenin. Blue regions superimposed on the monomeric crystallographic structure of lysenin in the “prepore” indicate peptides that were more and less stable in the lysenin prepore form than in its soluble form. Grey regions indicate peptides that were unchanged in the prepore form compared to the soluble form. In the “pore” structure: blue regions indicate regions that were more stable in the lysenin pore form than its prepore form; grey regions indicate peptides that were unchanged in the pore form compared to the prepore form. In the square the results overlaid on the protomer of the lysenin pore structure are shown.

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
