# Peer review of "Insight into the Structural Dynamics of the Lysenin During Prepore-to-Pore Transition Using Hydrogen–Deuterium Exchange Mass Spectrometry"

_toxins, 2019, doi:10.3390/toxins11080462_

Round 1
Reviewer 1 Report
This manuscript aims to gain insight into the structural dynamics of lysenin in the pre-pore and pore states, in an attempt to resolve the molecular mechanisms underlying this large conformational transition. To reach this goal, the authors compared lysenin-WT and a non-lytic mutant, lysenin-V88C/Y131C. Notably, the non-lytic mutant regains its lytic capability upon treatment with DTT, allowing direct comparison between the lytic and non-lytic form with the same protein preparation.Overall, the subject is of importance, the experiments are well-thought and executed and the conclusions are mostly sound. However, there are some issues that the authors should address.
Major points:
1. Section 2.2 describes the effect of V88C/Y181C mutation on oligomerization and pore-forming activity. The EM experiment clearly shows that the mutant forms an oligomeric assembly, but the oligomeric state could not be detected in the SDS-PAGE analysis performed in figure 4B, likely due to susceptibility to SDS treatment. Since this preparation of membrane-bound mutant is subsequently used for HDX-MS experiments, I think that a better characterization of the oligomeric state, which can affect and explain the HDX patterns, is required. The authors may consider using cross-linking prior to running the sample in the gel, for example, or using a native gel. This will provide important insights into the pre-pore state.
2. Given the central role of HDX-MS in this manuscript, I urge the authors to explain in the introduction what the method measures in a simple and precise manner. Currently, throughout the manuscript, the HDX is explained to rely on different factors. For example, in section 2.3, the degree of HDX is related only to the presence of secondary structure (lines 277-278), but in lines 295-297, it is related to solvent exposure. There is no need to go into the theory of HDX, but the reader should understand in a qualitative manner what affects the degree of HDX.
3. The presentation of the HDX-MS data is largely non-contributing to the reader. The plots are loaded with information, and the reader needs to go back and forth to understand which color belongs to which condition and to correlate the plots with the structure presented in Figure 1. It is hard for a non-expert in the field of lysenin to follow through this part of the results. Moreover, the actual data – the uptake plots as a function of time – are presented only for the aqueous form in Figure S3. I would suggest changing the presentation of the data as following:
(i) All the uptake plots should be presented in supplementary figures.
(ii) Figure 6: consider replacing panels A and B with the HDX pattern of WT lysenin at different time points presented as heat maps on the crystal structure. Label the structures with the secondary structure elements. Panels C and D can remain as they are, but I would narrow the y-scale to ±10% and add a legend to the panels to indicate the +/-DTT conditions.
(iii) Figure 7: Panels A-C can be replaced by heat maps plotted on the structures of the aqueous form + SM (for the mutant without DTT) or the membrane-embedded form (for the WT and WT with DTT). Consider showing the uptake also on the entire pore and then focusing on a single transmembrane monomer. Panel E can be divided into two panels or it should be labeled more clearly with an inset legend to indicate the +/-DTT conditions.
(iv) If the abovementioned changes are made, Figure 8 will be redundant and can be removed.
(v) Figure 9 – panels A,C show data presented in Figure 7 (20 min). I would suggest the authors to show only difference plots in this figure. All the time points can be shown on a single panel, with different times shown with different colors and clearly labeled. The y-scale should be adjusted to improve visibility of the results.
(vi) Figure 10 is a very good concluding figure. If the abovementioned suggestions will be accepted by the authors, this figure will consolidate the complex HDX part much better, in my opinion.
4. When discussing the pre-pore state, the authors should address the possible effect of SM binding (according to the previously described crystal structure) and the oligomeric state (if determined as suggested above) on the observed HDX patterns. This is especially important in light of the discussion regarding intermolecular salt bridges.
5. Careful should be made throughout the manuscript when using the term “stabilizing”. I suggest the authors to only address increase/decrease in HDX throughout the results section. In the discussion, they may suggest that decreased HDX reflects stabilization, but it is only one explanation to this observation. The reduction in solvent accessibility upon membrane insertion will reduce HDX, potentially even if the region is “destabilized”.
Minor points:
1. I suggest adding the structure of monomeric lysenin bound to SM to Figure 1. The binding of SM can potentially induce dynamic changes even without oligomerization, so it is worthwhile to illustrate the region involved, which are described in the introduction.
2. I would suggest using colors rather than line pattern in figures with multiple curves. This presentation is especially challenging to the reader in Figure 3 and 5B. I would also recommend adding an inset legend to Figure 5B as in Figure 3. As far as I understand, the use of color in figures does not entail additional costs.
3. There is a typo in Figure 3B – ‘wavelenght’ instead of ‘wavelength’.
4. The sequence coverage for each construct/condition should appear at the beginning of the results section describing the HDX-MS experiments. Figure S4 should precede Figure S3.
5. The panels presented in Figure 8 are not ordered according to the protein sequence. It is quite confusing.
6. Figure 12 can be expanded to show the proposed structure-dynamic model by using the same color-coding on the different states of lysenin. Maybe Figure 10 can replace this figure.
Reviewer 2 Report
This interesting paper describes structural studies aimed at elucidating conformational transitions from solution state to membrane bound state in the toxin Lysenin. The experimental part has been carefully carried out and the paper is very well written.
The authors used a cysteine double mutant (V88C/Y131C) that allowed redox modulation via disulfide bond formation to stabilize Lysenin and prevent pore forming transitions. The authors should mention that wild type Lysenin already contains one disulfide bond and should discuss the potential implications for their experimental design. For example, the necessary control experiment using wild type toxin + DTT has not always been conducted as far as I can see ?
Minor points:
Line 105: “although recent studies…” needs references.
Line 114: ”previous studies” needs a reference for the cysteine double mutant. Ref 23 mentioned in the next sentence doesn’t seem right.
Line 249/250: Check: “Hemolytic activity of lyseninWT (solid line), PFOV8C/Y131C (dashed line), and lyseninV88C/Y131C with 10 mM DTT (dotted line).” This doesn’t make sense and does not reflect what is mentioned in the text or shown in Fig 5b.
In fig 6C,D, please check. If D(WT) – D(Mutant) is shown (I assume this is %, please indicate), then wouldn't a positive value indicate that D(mutant) is smaller than D(WT) or the mutant is more protected than wt (i.e. stabilizing)? i.e. the opposite of what is shown?
Line 509: “Reduction the of disulfide”, swap ‘of’ and ‘the’
Figure 12: It is not entirely clear to me what this figure is showing. Are these two superimposed structures ? What is the difference between the blue and grey structural features?
In the experimental part the authors mention that the protein was produced as His-tagged fusion protein, but apparently the tag was not removed. I would like the authors to comment on whether this tag affects the activity of the toxin? In general it would be helpful to have the full amino acid sequence of the protein used in this study (incl. tags) included in the manuscript (or SI).
